# A New Quantum Private Protocol for Set Intersection Cardinality Based on a Quantum Homomorphic Encryption Scheme for Toffoli Gate

**DOI:** 10.3390/e25030516

**Published:** 2023-03-16

**Authors:** Wen Liu, Yangzhi Li, Zhirao Wang, Yugang Li

**Affiliations:** 1State Key Laboratory of Media Convergence and Communication, Communication University of China, Beijing 100024, China; 2School of Computer and Cyber Sciences, Communication University of China, Beijing 100024, China; 3Key Laboratory of Convergent Media and Intelligent Technology, Communication University of China, Ministry of Education, Beijing 100034, China; 4Academy of Broadcasting Science, Beijing 100045, China

**Keywords:** private set intersection cardinality, Pauli gates, Toffoli gate, quantum homomorphic encryption

## Abstract

Set Intersection Cardinality (SI-CA) computes the intersection cardinality of two parties’ sets, which has many important and practical applications such as data mining and data analysis. However, in the face of big data sets, it is difficult for two parties to execute the SI-CA protocol repeatedly. In order to reduce the execution pressure, a Private Set Intersection Cardinality (PSI-CA) protocol based on a quantum homomorphic encryption scheme for the Toffoli gate is proposed. Two parties encode their private sets into two quantum sequences and encrypt their sequences by way of a quantum homomorphic encryption scheme. After receiving the encrypted results, the semi-honest third party (TP) can determine the equality of two quantum sequences with the Toffoli gate and decrypted keys. The simulation of the quantum homomorphic encryption scheme for the Toffoli gate on two quantum bits is given by the IBM Quantum Experience platform. The simulation results show that the scheme can also realize the corresponding function on two quantum sequences.

## 1. Introduction

Secure multiparty computation (SMC) [1,2,3] is a crucial cryptographic primitive which fits the following description: Assume that there is a function typically specified by a map $F:{\left({\left\{0,1\right\}}^{*}\right)}^{n}\to {\left({\left\{0,1\right\}}^{*}\right)}^{n}$ and a set of *n* parties, $P=\{P_{1},...,P_{n}\}$, who want to compute values of this function with respect to their private data. Each party $P_{i}$ has its input $x_{i}\in {\left\{0,1\right\}}^{*}$ and output $y_{i}\in {\left\{0,1\right\}}^{*}$, following correspondence $y_{i}=F\left(x_{i}\right)$. Our target is to ensure that all parties in a subset C⊂P receive correct outputs from others while no information related to the input can be accessed. SMC has raised widespread concerns and has wide applications in electronic voting, cloud computing, online auction, etc.

A typical SMC [4] application is Private Set Intersection (PSI), which also known as Private Matching (PM). Specifically, PSI permits two parties, $P_{1}$ and $P_{2}$, who respectively have a private set $x_{1}$ and $x_{2}$. Without disclosing any information that does not belong to this intersection, they seek to find the intersection $x_{1}\cap x_{2}$. There have been many applications of PSI, such as privacy-preserving data mining [5], data outsourcing on cloud [6], location-based privacy-preserving sharing [7], testing of fully-sequenced human genomes [8], proximity testing [9], and other online services [10].

Due to the extensive and important applications, there have been many suggestions for PSI protocols. In 2004, Freedman et al. [4] first gave the definition of PSI and presented several PSI protocols by using homomorphic encryption and balanced hashing. Homomorphic encryption was first proposed by Rivest et al. in 1978 [11]. A new symmetric homomorphic functional encryption using modular multiplications over a hidden ring was proposed [12]. Then, some PSI protocols were proposed based on classical cryptography [13,14,15,16]. However, PSI reveals too much private information and it cannot meet the higher privacy requirements in some scenarios. In this case, Private Set Intersection Cardinality (PSI-CA) [17] was introduced, which can securely determine the size of set intersection and can be used to generate association rules. In [18], a PSI-CA protocol was the first to achieve security in the standard model under the Quadratic Residuosity QR assumption with linear complexities, which can hide the size of the client’s private set. In [19], a PSI-CA protocol was proposed, which had linear computation and communication complexities and was the most efficient PSI-CA protocol in previously proposed PSI-CA protocols [18,19]. PSI-CA only outputs the intersection cardinality and does not reveal the specific content of the intersection. The security of classical PSI-CA protocols is based on the computational complexity assumptions, which are vulnerable to attack by the quantum algorithms [20,21,22].

On the other hand, scholars began to seek a quantum approach to solving the PSI-CA problem. In [23], Shi et al. presented two quantum protocols to solve the Oblivious Set-member Decision problem. These protocols can be used to privately compute multi-party set intersection and union in the quantum domain. In [24], Shi et al. informally gave a definition of PSI first. Then they presented a quantum scheme for PSI based on *n* encoded states, *n* quantum operators, and *n* von Neumann measurements. In [25], Arpita gave a two-party protocol for computing set intersection securely in the quantum domain in a rational setting, where the players are trying to maximize their utilities. However, PSI reveals too much private personal information in some scenarios. In order to prevent revealing the specific content, Shi et al. proposed some quantum protocols of PSI-CA [26,27,28]. PSI-CA and PSU-CA enable two parties, each with a private set, to jointly compute the cardinality of their intersection or union without disclosing any private information about their respective sets. These protocols are useful in social networks and for privacy-preserving data mining.

In this paper, following the idea in [26], we propose a PSI-CA protocol based on a quantum homomorphic encryption scheme for the Toffoli gate. With the help of a semi-honest TP, two parties can use this protocol to privately obtain the number of all their private sets’ common elements. When the amount of data is large, two parties, which do not have strong quantum computing capabilities, only prepare and encrypt quantum single-particle states. The role of semi-honest TP is to execute the protocol loyally and record all the results of its intermediate computations. However, the TP cannot learn anything about the private information. In our protocol, the semi-honest third party (TP) can be used to perform Toffoli gate and decryption operations. It will keep a record of all its intermediate results and might try to infer the private inputs from the record. Our protocol is simpler and easier to implement.

This paper is organized as follows: we introduce some correlative preliminaries in Section 2; we propose a quantum PSI-CA protocol in Section 3; in Section 4, we analyze the correctness and security of our protocol and describe the implementation of our protocols on the IBM Quantum Experience platform. A brief discussion and the concluding summary are given in Section 5.

## 2. Preliminary

### 2.1. Pauli Gates

Some operators are introduced first. Four single-qubit operators I,X,Y,Z are shown as follows:(1)$$I=\left[\begin{array}{cc} 1 & 0\\ 0 & 1 \end{array}\right],\;\;X=\left[\begin{array}{cc} 0 & 1\\ 1 & 0 \end{array}\right],\;\;Y=\left[\begin{array}{cc} 0 & -i\\ i & 0 \end{array}\right],\;\;Z=\left[\begin{array}{cc} 1 & 0\\ 0 & -1 \end{array}\right]$$
The circuit symbols for the four single-qubit gates I,X,Y,Z are shown in Figure 1.

### 2.2. Quantum Toffoli Gate

The quantum Toffoli gate (called the *T* gate) is seen as an important component in the theory of quantum computation. The unitary transform matrix of the *T* gate is as follows:(2)$$T=\left[\begin{array}{cccccccc} 1 & 0 & 0 & 0 & 0 & 0 & 0 & 0\\ 0 & 1 & 0 & 0 & 0 & 0 & 0 & 0\\ 0 & 0 & 1 & 0 & 0 & 0 & 0 & 0\\ 0 & 0 & 0 & 1 & 0 & 0 & 0 & 0\\ 0 & 0 & 0 & 0 & 1 & 0 & 0 & 0\\ 0 & 0 & 0 & 0 & 0 & 1 & 0 & 0\\ 0 & 0 & 0 & 0 & 0 & 0 & 0 & 1\\ 0 & 0 & 0 & 0 & 0 & 0 & 1 & 0 \end{array}\right]$$

The *T* gate has three input bits and three output bits. For a three-qubit quantum system, $|a\rangle |b\rangle |c\rangle$, the quantum *T* gate will act as:(3)$$T|a\rangle |b\rangle |c\rangle =|a\rangle |b\rangle |c⊕(a\cdot b)\rangle .$$

The circuit symbol for the *T* gate is shown in Figure 2.

### 2.3. Information-Theoretic Security

In [23], the conception of mixed states is introduced and a quantum information-theoretic security criterion for a quantum protocol is given as follows:

The protocol is informationally secure for every input state $\varphi_{in}$ if the output state $\varphi_{out}$ is the totally mixed state. The relation of the input state $\varphi_{in}$ and the output state $\varphi_{out}$ is as follows:(4)$$\varphi_{out}=\sum_{k}\frac{1}{2^{2n}}U_{k}\varphi_{in}{\left(U_{k}^{}\right)}^{†}=\frac{1}{2^{n}}I_{2^{n}},$$
where $\varphi_{in}$ is the density operator of all possible *n*-qubit input states and $U_{k}$ are the corresponding unitary operations applied on input state.

## 3. Quantum Private Computation Protocol for Set Intersection Cardinality

We use the definition of PSI-CA [19]. Suppose that there are two parties, Alice and Bob. They input a private set $S_{A}=\left\{a_{1},a_{2},...,a_{n_{1}}\right\}$ and $S_{B}=\left\{b_{1},b_{2},...,b_{n_{2}}\right\}$, respectively. *S* is a complete set $\left\{x_{1},x_{2},...,x_{n}\right\}$ and $S_{A},S_{B}\subset S$. After running the PSI-CA protocol with a help of the semi-honest third party, Calvin, Alice and Bob output the cardinality of the intersection of their private sets, i.e., $\left|S_{A}\cap S_{B}\right|$, without leaking any information about their sets. The quantum scheme for PSI-CA is described as follows:

(1) Alice and Bob each prepare a $\left(n+n^{\prime }\right)$-photon sequence, denoted by $Sq_{A}=\left(|\psi_{1}^{A}\rangle ,|\psi_{2}^{A}\rangle ,...,|\psi_{n+n{}^{\prime }}^{A}\rangle \right),Sq_{B}=\left(|\psi_{1}^{B}\rangle |\psi_{2}^{B}\rangle ,...,|\psi_{n+n{}^{\prime }}^{B}\rangle \right)$. The first *n* particles of $Sq_{A},Sq_{B}$ are prepared according to Alice’s and Bob’s private sets $S_{A},S_{B}$:(5)$$\left\{\begin{array}{c} \;|\psi_{i}^{A}\rangle =|1\rangle ,\;if\;x_{i}\in S_{A}\;\;\;\\ |\psi_{i}^{A}\rangle =|0\rangle ,\;if\;x_{i}\notin S_{A}\;\;\;\;\;\;\; \end{array}\;\left\{\begin{array}{c} \;|\psi_{i}^{B}\rangle =|1\rangle ,\;if\;x_{i}\in S_{B}\;\;\;\\ |\psi_{i}^{B}\rangle =|0\rangle ,\;if\;x_{i}\notin S_{B}\;\;\;\;\;\;\; \end{array}\right.\right.$$
The last $n^{\prime }$ particles of $Sq_{A},Sq_{B}$ are dummy photons, which are randomly chosen from $\left\{|0\rangle ,|1\rangle \right\}$.

(2) Alice and Bob work together to find the number of $|\psi_{i}^{A}\rangle =|\psi_{i}^{B}\rangle =|1\rangle \left(i=n+1,...,n+n^{{}^{\prime }}\right)$, denoted by $N_{CA}^{{}^{\prime }}$, which means how many bits are equal and equal to $|1\rangle$ in the last $n^{\prime }$ particles of $Sq_{A},Sq_{B}$.

They also permutate $Sq_{A},Sq_{B}$ using the same permutation regulation π. The new sequences are denoted by $Sq_{A}^{{}^{\prime }}=\left(|\psi_{1}^{A^{\prime }}\rangle ,|\psi_{2}^{A^{\prime }}\rangle ,...,|\psi_{n+n{}^{\prime }}^{A^{\prime }}\rangle \right),Sq_{B}^{{}^{\prime }}=\left(|\psi_{1}^{B^{\prime }}\rangle ,|\psi_{2}^{B^{\prime }}\rangle ,...,|\psi_{n+n{}^{\prime }}^{B^{\prime }}\rangle \right)$.

Each of them chooses a sequence, $L_{A}=(l_{1}^{A},l_{2}^{A},l_{3}^{A},l_{4}^{A},...,l_{2(n+n^{\prime })-1}^{A},$ $l_{2(n+n^{\prime })}^{A})\left(L_{B}=\left(l_{1}^{B},l_{2}^{B},l_{3}^{B},l_{4}^{B},...,l_{2(n+n^{\prime })-1}^{B},l_{2(n+n^{\prime })}^{B}\right)\right)$, where $l_{2k-1}^{B},l_{2k}^{B}$ are randomly chosen from {0,1}. Then, she(he) uses the Quantum One-time Pad algorithm (QOTP) [25] to encrypt the *k*th particle of $Sq_{A}^{{}^{\prime }}\left(Sq_{B}^{{}^{\prime }}\right)$ and get $Z^{l_{2k-1}^{A}}X^{l_{2k}^{A}}|\psi_{k}^{A}\rangle$ $\left(Z^{l_{2k-1}^{B}}X^{l_{2k}^{B}}|\psi_{k}^{B}\rangle \right)$. The new particles sequence is denoted by $S_{A}^{{}^{\prime \prime }}=(Z^{l_{1}^{A}}X^{l_{2}^{A}}|\psi_{1}^{A^{\prime }}\rangle ,$ $...,Z^{l_{2(n+n^{\prime })-1}^{A}}X^{l_{2(n+n^{\prime })}^{A}}|\psi_{n+n{}^{\prime }}^{A^{\prime }}\rangle )(S_{B}^{{}^{\prime \prime }}=(Z^{l_{1}^{B}}X^{l_{2}^{B}}|\psi_{1}^{B^{\prime }}\rangle ,$$...,Z^{l_{2(n+n^{\prime })-1}^{B}}X^{l_{2(n+n^{\prime })}^{B}}$$|\psi_{n+n{}^{\prime }}^{B^{\prime }}\rangle ))$.

Alice (Bob) also inserts some checking particles, which are randomly chosen from $\{|0\rangle ,|1\rangle ,$ $|+\rangle ,|-\rangle \}$, into $S_{A}^{{}^{\prime \prime }}\left(S_{B}^{{}^{\prime \prime }}\right)$ and sends the new sequence $S_{A}^{{}^{\prime \prime \prime }}\left(S_{B}^{{}^{\prime \prime \prime }}\right)$ to the third party Calvin.

After that, Alice(Bob) transmits the insert positions $Po_{A}\left(Po_{B}\right)$ and $L_{A}\left(L_{B}\right)$ to Calvin using the quantum secure direct communication (QSDC) protocol. QSDC is one of the most important branches of quantum communication and it directly transmits secret messages.

(3) After receiving $S_{A}^{{}^{\prime \prime \prime }},S_{B}^{{}^{\prime \prime \prime }}$, Alice, Bob, and Calvin perform the eavesdropping check using the insert positions $Po_{A},Po_{B}$ and the measuring bases of checking particles. If the error rate exceeds the threshold they preset, they abort the scheme. Otherwise, they discard the measured photons in $S_{A}^{{}^{\prime \prime \prime }},S_{B}^{{}^{\prime \prime \prime }}$ and Calvin gets two sequences $S_{A}^{{}^{\prime \prime }}=\left(Z^{l_{1}^{A}}X^{l_{2}^{A}}|\psi_{1}^{A^{\prime }}\rangle ,...,Z^{l_{2(n+n^{\prime })-1}^{A}}X^{l_{2(n+n^{\prime })}^{A}}|\psi_{(}n+n^{\prime }{)}^{A^{\prime }}\rangle \right),S_{B}^{{}^{\prime \prime }}=(Z^{l_{1}^{B}}X^{l_{2}^{B}}|\psi_{1}^{B^{\prime }}\rangle ,...,Z^{l_{2(n+n^{\prime })-1}^{B}}X^{l_{2(n+n^{\prime })}^{B}}$$|\psi_{(}n+n^{\prime }{)}^{B^{\prime }}\rangle )$.

Calvin prepares a sequence $S_{C}=\left(|\psi_{1}^{C}\rangle ,|\psi_{2}^{C}\rangle ,...,|\psi_{n+n^{\prime }}^{C}\rangle \right)$, where $|\psi_{i}^{C}\rangle$ is randomly chosen from $\left\{|0\rangle ,|1\rangle \right\}$.

(4) Calvin executes some operations on the *i*th quantum bits of $S_{A}^{{}^{\prime \prime }},S_{B}^{{}^{\prime \prime }},S_{C}$ and gets:(6)$$\begin{array}{c} {|\psi_{i}^{A^{\prime \prime }}\rangle }_{1}{|\psi_{i}^{B^{\prime \prime }}\rangle }_{2}{|\psi_{i}^{C^{\prime }}\rangle }_{3}\\ =\left(CNOT_{1,3}^{l_{2i}^{B}}⊗I_{2}\right)\left(I_{1}⊗CNOT_{2,3}^{l_{2i}^{A}}\right)\left(Z^{l_{2i-1}^{A}}X^{l_{2i}^{A}}⊗Z^{l_{2i-1}^{B}}X^{l_{2i}^{B}}⊗X^{l_{2i}^{A}l_{2i}^{B}}\right)T\left(Z^{l_{2i-1}^{A}}X^{l_{2i}^{A}}⊗Z^{l_{2i-1}^{B}}X^{l_{2i}^{B}}⊗I\right){|\psi_{i}^{A^{\prime }}\rangle }_{1}{|\psi_{i}^{B^{\prime }}\rangle }_{2}{|\psi_{i}^{C}\rangle }_{3}\\ ={|\psi_{i}^{A^{\prime }}\rangle }_{1}{|\psi_{i}^{B^{\prime }}\rangle }_{2}{|\psi_{i}^{C}⊕\psi_{i}^{A^{\prime }}\psi_{i}^{B^{\prime }}\rangle }_{3}. \end{array}$$

Calvin measures $|\psi_{i}^{C^{\prime }}\rangle$ using the *X* basis and compares the measurement result with $|\psi_{i}^{C}\rangle$. He also counts how many quantum bits $|\psi_{i}^{C^{\prime }}\rangle ,|\psi_{i}^{C}\rangle$ are different and the number is denoted by $N_{CA^{\prime \prime }}$. It is obvious that the intersection cardinality of $S_{A},S_{B}$ is equal to $N_{CA^{\prime \prime }}-N_{CA^{\prime }}$.

We have to point out that if Alice and Bob apply a NOT gate on each particle of $Sq_{A},Sq_{B}$ in step(1), the private set union cardinality of $S_{A},S_{B}$ is equal to $\left|S|-(N_{CA^{\prime \prime }}-N_{CA^{\prime }}\right)$ using the PSI-CA quantum protocol.

## 4. Analysis and Comparison

### 4.1. Correctness Analysis

In this section, we illustrate the correctness of our protocol. Figure 3 describes the circuit *U* used to privately apply the *T* gate on $|\psi_{i}^{A}\rangle |\psi_{i}^{B}\rangle |\psi_{i}^{C}\rangle$, where $l_{2i-1}^{A},l_{2i}^{A},l_{2i-1}^{B},l_{2i}^{B}\in \left\{0,1\right\}$. For $i=1,2,...,n+n^{\prime }$, Alice, Bob and Calvin can use the circuit *U* to privately calculate $T|\psi_{i}^{A}\rangle |\psi_{i}^{B}\rangle |\psi_{i}^{C}\rangle$. If $|\psi_{i}^{C}\rangle$ is reversed, they can determine $|\psi_{k}^{A}\rangle =|\psi_{k}^{B}\rangle =|1\rangle$.

According to the circuit *U*, it can be verified that
(7)$$\begin{array}{c} \left(CNOT_{1,3}^{0}⊗I_{2}\right)\left(I_{1}⊗CNOT_{2,3}^{0}\right)\left(Z^{0}X^{0}⊗Z^{0}X^{0}⊗X^{0}\right)T\left(Z^{0}X^{0}⊗Z^{0}X^{0}⊗I\right)|\psi_{i}^{A}\rangle |\psi_{i}^{B}\rangle |\psi_{i}^{C}\rangle \\ =T|\psi_{i}^{A}\rangle |\psi_{i}^{B}\rangle |\psi_{i}^{C}\rangle . \end{array}$$
(8)$$\begin{array}{c} \left(CNOT_{1,3}^{0}⊗I_{2}\right)\left(I_{1}⊗CNOT_{2,3}^{0}\right)\left(Z^{0}X^{0}⊗Z^{1}X^{0}⊗X^{0}\right)T\left(Z^{0}X^{0}⊗Z^{1}X^{0}⊗I\right)|\psi_{i}^{A}\rangle |\psi_{i}^{B}\rangle |\psi_{i}^{C}\rangle \\ =T|\psi_{i}^{A}\rangle |\psi_{i}^{B}\rangle |\psi_{i}^{C}\rangle . \end{array}$$
(9)$$\begin{array}{c} \left(CNOT_{1,3}^{0}⊗I_{2}\right)\left(I_{1}⊗CNOT_{2,3}^{0}\right)\left(Z^{1}X^{0}⊗Z^{0}X^{0}⊗X^{0}\right)T\left(Z^{1}X^{0}⊗Z^{0}X^{0}⊗I\right)|\psi_{i}^{A}\rangle |\psi_{i}^{B}\rangle |\psi_{i}^{C}\rangle \\ =T|\psi_{i}^{A}\rangle |\psi_{i}^{B}\rangle |\psi_{i}^{C}\rangle \end{array}$$
(10)$$\begin{array}{c} \left(CNOT_{1,3}^{0}⊗I_{2}\right)\left(I_{1}⊗CNOT_{2,3}^{0}\right)\left(Z^{1}X^{0}⊗Z^{1}X^{0}⊗X^{0}\right)T\left(Z^{1}X^{0}⊗Z^{1}X^{0}⊗I\right)|\psi_{i}^{A}\rangle |\psi_{i}^{B}\rangle |\psi_{i}^{C}\rangle \\ =T|\psi_{i}^{A}\rangle |\psi_{i}^{B}\rangle |\psi_{i}^{C}\rangle . \end{array}$$
(11)$$\begin{array}{c} \left(CNOT_{1,3}^{0}⊗I_{2}\right)\left(I_{1}⊗CNOT_{2,3}^{1}\right)\left(Z^{0}X^{1}⊗Z^{0}X^{0}⊗X^{0}\right)T\left(Z^{0}X^{1}⊗Z^{0}X^{0}⊗I\right)|\psi_{i}^{A}\rangle |\psi_{i}^{B}\rangle |\psi_{i}^{C}\rangle \\ =T|\psi_{i}^{A}\rangle |\psi_{i}^{B^{\prime }}\rangle |\psi_{i}^{C}\rangle \end{array}$$
(12)$$\begin{array}{c} \left(CNOT_{1,3}^{0}⊗I_{2}\right)\left(I_{1}⊗CNOT_{2,3}^{1}\right)\left(Z^{0}X^{1}⊗Z^{1}X^{0}⊗X^{0}\right)T\left(Z^{0}X^{1}⊗Z^{1}X^{0}⊗I\right)|\psi_{i}^{A}\rangle |\psi_{i}^{B}\rangle |\psi_{i}^{C}\rangle \\ =T|\psi_{i}^{A}\rangle |\psi_{i}^{B}\rangle |\psi_{i}^{C}\rangle \end{array}$$
(13)$$\begin{array}{c} \left(CNOT_{1,3}^{0}⊗I_{2}\right)\left(I_{1}⊗CNOT_{2,3}^{1}\right)\left(Z^{1}X^{1}⊗Z^{0}X^{0}⊗X^{0}\right)T\left(Z^{1}X^{1}⊗Z^{0}X^{0}⊗I\right)|\psi_{i}^{A}\rangle |\psi_{i}^{B}\rangle |\psi_{i}^{C}\rangle \\ =-T|\psi_{i}^{A}\rangle |\psi_{i}^{B}\rangle |\psi_{i}^{C}\rangle \end{array}$$
(14)$$\begin{array}{c} \left(CNOT_{1,3}^{0}⊗I_{2}\right)\left(I_{1}⊗CNOT_{2,3}^{1}\right)\left(Z^{1}X^{1}⊗Z^{1}X^{0}⊗X^{0}\right)T\left(Z^{1}X^{1}⊗Z^{1}X^{0}⊗I\right)|\psi_{i}^{A}\rangle |\psi_{i}^{B}\rangle |\psi_{i}^{C}\rangle \\ =-T|\psi_{i}^{A}\rangle |\psi_{i}^{B}\rangle |\psi_{i}^{C}\rangle \end{array}$$
(15)$$\begin{array}{c} \left(CNOT_{1,3}^{1}⊗I_{2}\right)\left(I_{1}⊗CNOT_{2,3}^{0}\right)\left(Z^{0}X^{0}⊗Z^{0}X^{1}⊗X^{0}\right)T\left(Z^{0}X^{0}⊗Z^{0}X^{1}⊗I\right)|\psi_{i}^{A}\rangle |\psi_{i}^{B}\rangle |\psi_{i}^{C}\rangle \\ =-T|\psi_{i}^{A}\rangle |\psi_{i}^{B}\rangle |\psi_{i}^{C}\rangle \end{array}$$
(16)$$\begin{array}{c} \left(CNOT_{1,3}^{1}⊗I_{2}\right)\left(I_{1}⊗CNOT_{2,3}^{0}\right)\left(Z^{0}X^{0}⊗Z^{1}X^{1}⊗X^{0}\right)T\left(Z^{0}X^{0}⊗Z^{1}X^{1}⊗I\right)|\psi_{i}^{A}\rangle |\psi_{i}^{B}\rangle |\psi_{i}^{C}\rangle \\ =-T|\psi_{i}^{A}\rangle |\psi_{i}^{B}\rangle |\psi_{i}^{C}\rangle \end{array}$$
(17)$$\begin{array}{c} \left(CNOT_{1,3}^{1}⊗I_{2}\right)\left(I_{1}⊗CNOT_{2,3}^{0}\right)\left(Z^{1}X^{0}⊗Z^{0}X^{1}⊗X^{0}\right)T\left(Z^{1}X^{0}⊗Z^{0}X^{1}⊗I\right)|\psi_{i}^{A}\rangle |\psi_{i}^{B}\rangle |\psi_{i}^{C}\rangle \\ =-T|\psi_{i}^{A}\rangle |\psi_{i}^{B}\rangle |\psi_{i}^{C}\rangle \end{array}$$
(18)$$\begin{array}{c} \left(CNOT_{1,3}^{1}⊗I_{2}\right)\left(I_{1}⊗CNOT_{2,3}^{0}\right)\left(Z^{1}X^{0}⊗Z^{1}X^{1}⊗X^{0}\right)T\left(Z^{1}X^{0}⊗Z^{1}X^{1}⊗I\right)|\psi_{i}^{A}\rangle |\psi_{i}^{B}\rangle |\psi_{i}^{C}\rangle \\ =-T|\psi_{i}^{A}\rangle |\psi_{i}^{B}\rangle |\psi_{i}^{C}\rangle \end{array}$$
(19)$$\begin{array}{c} \left(CNOT_{1,3}^{1}⊗I_{2}\right)\left(I_{1}⊗CNOT_{2,3}^{1}\right)\left(Z^{0}X^{1}⊗Z^{0}X^{1}⊗X^{1}\right)T\left(Z^{0}X^{1}⊗Z^{0}X^{1}⊗I\right)|\psi_{i}^{A}\rangle |\psi_{i}^{B^{\prime }}\rangle |\psi_{i}^{C}\rangle \\ =T|\psi_{i}^{A}\rangle |\psi_{i}^{B}\rangle |\psi_{i}^{C}\rangle \end{array}$$
(20)$$\begin{array}{c} \left(CNOT_{1,3}^{1}⊗I_{2}\right)\left(I_{1}⊗CNOT_{2,3}^{1}\right)\left(Z^{0}X^{1}⊗Z^{1}X^{1}⊗X^{1}\right)T\left(Z^{0}X^{1}⊗Z^{1}X^{1}⊗I\right)|\psi_{i}^{A}\rangle |\psi_{i}^{B}\rangle |\psi_{i}^{C}\rangle \\ =-T|\psi_{i}^{A}\rangle |\psi_{i}^{B}\rangle |\psi_{i}^{C}\rangle \end{array}$$
(21)$$\begin{array}{c} \left(CNOT_{1,3}^{1}⊗I_{2}\right)\left(I_{1}⊗CNOT_{2,3}^{1}\right)\left(Z^{1}X^{1}⊗Z^{0}X^{1}⊗X^{1}\right)T\left(Z^{1}X^{1}⊗Z^{0}X^{1}⊗I\right)|\psi_{i}^{A}\rangle |\psi_{i}^{B}\rangle |\psi_{i}^{C}\rangle \\ =-T|\psi_{i}^{A}\rangle |\psi_{i}^{B}\rangle |\psi_{i}^{C}\rangle \end{array}$$
(22)$$\begin{array}{c} \left(CNOT_{1,3}^{1}⊗I_{2}\right)\left(I_{1}⊗CNOT_{2,3}^{1}\right)\left(Z^{1}X^{1}⊗Z^{1}X^{1}⊗X^{1}\right)T\left(Z^{1}X^{1}⊗Z^{1}X^{1}⊗I\right)|\psi_{i}^{A}\rangle |\psi_{i}^{B}\rangle |\psi_{i}^{C}\rangle \\ =T|\psi_{i}^{A}\rangle |\psi_{i}^{B}\rangle |\psi_{i}^{C}\rangle . \end{array}$$

According to Equations (7)–(22), we can obtain
(23)$$\begin{array}{c} \left(CNOT_{1,3}^{l_{2i}^{B}}⊗I_{2}\right)\left(I_{1}⊗CNOT_{2,3}^{l_{2i}^{A}}\right)\left(Z^{l_{2i-1}^{A}}X^{l_{2i}^{A}}⊗Z^{l_{2i-1}^{B}}X^{l_{2i}^{B}}⊗X^{l_{2i}^{A}l_{2i}^{B}}\right)T\left(Z^{l_{2i-1}^{A}}X^{l_{2i}^{A}}⊗Z^{l_{2i-1}^{B}}X^{l_{2i}^{B}}⊗I\right)|\psi_{i}^{A}\rangle |\psi_{i}^{B}\rangle |\psi_{i}^{C}\rangle \\ =T|\psi_{i}^{A}\rangle |\psi_{i}^{B}\rangle |\psi_{i}^{C}\rangle \\ =|\psi_{i}^{A}\rangle |\psi_{i}^{B}\rangle |\psi_{i}^{C}⊕\left(\psi_{i}^{A}\cdot \psi_{i}^{B}\right)\rangle . \end{array}$$

Calvin measures $|\psi_{i}^{C}⊕\left(\psi_{i}^{A^{{}^{\prime }}}\cdot \psi_{i}^{B^{{}^{\prime }}}\right)\rangle$. If $|\psi_{i}^{C}⊕\left(\psi_{i}^{A^{{}^{\prime }}}\cdot \psi_{i}^{B^{{}^{\prime }}}\right)\rangle$ is different from $|\psi_{i}^{C}\rangle$, we can know $|\psi_{i}^{A^{{}^{\prime }}}\rangle =|\psi_{i}^{B^{{}^{\prime }}}\rangle =|1\rangle$. Alice and Bob have a common element in $S_{A},S_{B}$.

Suppose that the private set of Alice is $S_{A}=\left\{2,4\right\}$ and the private set of Bob is $S_{B}=\left\{3,4\right\}$ where a complete set is S={2,3,4}. The photon sequence of Alice is $S_{q_{A}}=\left\{|1\rangle ,|0\rangle ,|1\rangle \right\}$ and the photon sequence of Bob is $S_{q_{B}}=\left\{|0\rangle ,|1\rangle ,|1\rangle \right\}$. Calvin prepares a sequence $S_{q_{C}}=\left\{|1\rangle ,|0\rangle ,|0\rangle \right\}$. Alice chooses a sequence $L_{A}=\left(0,1,1,1,1,0\right)$ and Bob chooses a sequence $L_{B}=\left(0,0,1,1,0,1\right)$. Alice, Bob and Calvin perform some operations on $\left\{|1\rangle ,|0\rangle ,|1\rangle \right\},\left\{|0\rangle ,|1\rangle ,|1\rangle \right\},\left\{|1\rangle ,|0\rangle ,|0\rangle \right\}$ using $L_{A},L_{B}$ and get $\left(CNOT_{1,3}^{0}⊗I_{2}\right)\left(I_{1}⊗CNOT_{2,3}^{1}\right)\left(Z^{0}X^{1}⊗Z^{0}X^{0}⊗X^{0}\right)T\left(Z^{0}X^{1}⊗Z^{0}X^{0}⊗I\right)|1\rangle |0\rangle |1\rangle ,\left(CNOT_{1,3}^{1}⊗I_{2}\right)\left(I_{1}⊗CNOT_{2,3}^{1}\right)\left(Z^{1}X^{1}⊗Z^{1}X^{1}⊗X^{1}\right)T\left(Z^{1}X^{1}⊗Z^{1}X^{1}⊗I\right)|0\rangle |1\rangle |0\rangle ,$$\left(CNOT_{1,3}^{1}⊗I_{2}\right)\left(I_{1}⊗CNOT_{2,3}^{0}\right)$$\left(Z^{1}X^{0}⊗Z^{0}X^{1}⊗X^{0}\right)T\left(Z^{1}X^{0}⊗Z^{0}X^{1}⊗I\right)|1\rangle |1\rangle |0\rangle$. Then they can get $T\left(|1\rangle |0\rangle |1\rangle \right),T(|0\rangle$$|1\rangle |0\rangle ),T\left(|1\rangle |1\rangle |0\rangle \right)$ and the new photon sequence of Calvin is $|1⊕(1\cdot 0)\rangle |0⊕(0\cdot 1)\rangle$$|0⊕(1\cdot 1)\rangle$. Only the third photon in Calvin’s new sequence $|0⊕(1\cdot 1)\rangle =|1\rangle \}$ is different from the third photon of his original sequence $|0\rangle \}$. So we can get that Alice and Bob have only one common element in $S_{A},S_{B}$.

### 4.2. Implementation of Quantum PSI-CA Protocols on IBM Quantum Experience Platform

Now, we move forward through a similar approach to experimentally realize our PSI-CA protocol on the IBM Quantum Experience platform. Let us say the two parties, Alice and Bob, have a private set $S_{A}$ and $S_{B}$, respectively, where *S* is a complete set and $S_{A},S_{B}\in S$. For the encoding procedure, $S_{A}$ and $S_{B}$ are encoded into two $\left(n+n^{\prime }\right)$-particle sequences. Alice, Bob, and Calvin can privately apply the *T* gate on their corresponding position particles using the IBM Quantum Experience platform. The measuring results of Calvin’s particle are related to the PSI-CA of $S_{A},S_{B}$.

The circuit on the IBM Quantum Experience platform for privately computing for eight cases of $T|\psi_{A0}\rangle |\psi_{B0}\rangle |\psi_{C0}\rangle$ and the experiment results with 1024 shots for eight cases on the quantum circuit are shown in Figure 4, Figure 5, Figure 6, Figure 7, Figure 8, Figure 9, Figure 10 and Figure 11. In the experiment results’ figures, the x-axis represents 16 measurement results, and each of them includes the $T|\psi_{A0}\rangle |\psi_{B0}\rangle |\psi_{C0}\rangle$ and the information of $l_{A0},l_{A1},l_{B0},l_{B1}$.The y-axis represents the frequency of each measurement result. The first three binary bits in the x-axis correspond to the output of $T|\psi_{A0}\rangle |\psi_{B0}\rangle |\psi_{C0}\rangle$ and the following four binary bits in the x-axis are $l_{A0},l_{A1},l_{B0},l_{B2}$.

In Figure 4, $|\psi_{A0}\rangle =|1\rangle ,|\psi_{B0}\rangle =|1\rangle ,|\psi_{C0}\rangle =|1\rangle$. Take the measurement results “1101010”, for example, in Figure 4, the last four bits 1010 represent the measurement results of $l_{A0},l_{A1},l_{B0},l_{B1}$, which are used to control the gates in the quantum circuit. The first three bits 110 represent the new measurement result of $|\psi_{A0}\rangle ,|\psi_{B0}\rangle ,|\psi_{C0}\rangle$ after operating the gates in the quantum circuit. From the frequency of each measurement result in Figure 4, it can be verified that no matter what the $l_{A0},l_{A1},l_{B0},l_{B1}$ is, the circuit will act as a *T* gate on $|\psi_{A0}\rangle =|1\rangle ,|\psi_{B0}\rangle =|1\rangle ,|\psi_{C0}\rangle =|1\rangle$. Using the same analysis method, we can reach the same conclusion from the frequency of each measurement result in Figure 5, Figure 6, Figure 7, Figure 8, Figure 9, Figure 10 and Figure 11.

### 4.3. Security Analysis

In this section, we verify the security of our quantum PSI-CA scheme by analyzing an external outside attack and a participant attack, respectively.

#### 4.3.1. Outside Attacks

In terms of outside attacks, this protocol allows for outside eavesdroppers to attack the quantum channel and obtain Alice and Bob’s particle sequences in step (2). Checking particles are introduced to to defend against it. With several checking particles inserted, the security checking procedure in Step (3) can detect the intercept–resend attack, the measurement–resend attack, the entanglement–measure attack, and the denial-of-service (DOS) attack with a nonzero probability.

In addition to this naive attack, there are some special forms of attack such as the delay photon Trojan horse attack, the invisible photon eavesdropping (IPE) Trojan horse attack, and the photon-number-splitting (PNS) attack, which are also available to outside eavesdroppers. In response to these attacks, we use several defenses. To defeat the delay-photon Trojan horse attack, we can use a photon-number splitter. To defeat the IPE attack, we can insert filters in front of their devices to filter out the photon signal with an illegitimate wavelength. To defeat the PNS attack, we can use the technology of beam splitters to split the sampling signals and judge whether these received photons are single photons or multiple photons. Therefore, the outside attacks are invalid to our protocol.

#### 4.3.2. Participant Attack

Gao et al. proposed the term “participant attack” in Ref. [29], which has attracted much attention in the cryptanalysis of quantum cryptography. It underlines that malicious user attacks are typically more potent and should be given more consideration. We analyze the possibility that Alice, Bob, and Calvin could use participant attacks to learn knowledge about the private binary strings in our protocol. Since both Alice and Bob’s sequences are sent to Calvin after processing, it is most critical to consider Calvin’s behavior.

In our protocol, Calvin only gets two-particle sequences $S_{A}^{{}^{\prime \prime }},S_{B}^{{}^{\prime \prime }}$. Calvin applies the *T* gate on each sequence in step (3).

According to the definition of information-theoretic security given in Section 2.3, we can know that the output state of step (2) in our protocol can be described as follows:(24)$$\begin{array}{c} \frac{1}{2^{2}}\sum_{l_{2i-1}^{A},l_{2i}^{A}\in \left\{0,1\right\}}Z^{l_{2i-1}^{A}}X^{l_{2i}^{A}}|\psi_{i}^{A}\rangle {\left(Z^{l_{2i-1}^{A}}X^{l_{2i}^{A}}\right)}^{†}\\ =\frac{1}{4}Z^{0}X^{0}\left(\frac{1}{2}|0\rangle \langle 0|+\frac{1}{2}|1\rangle \langle 1|\right){\left(Z^{0}X^{0}\right)}^{†}+\frac{1}{4}Z^{0}X^{1}\left(\frac{1}{2}|0\rangle \langle 0|+\frac{1}{2}|1\rangle \langle 1|\right){\left(Z^{0}X^{1}\right)}^{†}\left[+2ex\right]\;\;\;\\ +\frac{1}{4}Z^{1}X^{0}\left(\frac{1}{2}|0\rangle \langle 0|+\frac{1}{2}|1\rangle \langle 1|\right){\left(Z^{1}X^{0}\right)}^{†}+\frac{1}{4}Z^{1}X^{1}\left(\frac{1}{2}|0\rangle \langle 0|+\frac{1}{2}|1\rangle \langle 1|\right){\left(Z^{1}X^{1}\right)}^{†}\left[+2ex\right]\\ =\frac{1}{2}\left[\begin{array}{cc} 1 & 0\\ 0 & 1 \end{array}\right] \end{array}$$
(25)$$\begin{array}{c} \frac{1}{2^{2}}\sum_{l_{2i-1}^{B},l_{2i}^{B}\in \left\{0,1\right\}}Z^{l_{2i-1}^{B}}X^{l_{2i}^{B}}|\psi_{i}^{B}\rangle {\left(Z^{l_{2i-1}^{B}}X^{l_{2i}^{B}}\right)}^{†}\\ =\frac{1}{4}Z^{0}X^{0}\left(\frac{1}{2}|0\rangle \langle 0|+\frac{1}{2}|1\rangle \langle 1|\right){\left(Z^{0}X^{0}\right)}^{†}+\frac{1}{4}Z^{0}X^{1}\left(\frac{1}{2}|0\rangle \langle 0|+\frac{1}{2}|1\rangle \langle 1|\right){\left(Z^{0}X^{1}\right)}^{†}\\ \;\;\;+\frac{1}{4}Z^{1}X^{0}\left(\frac{1}{2}|0\rangle \langle 0|+\frac{1}{2}|1\rangle \langle 1|\right){\left(Z^{1}X^{0}\right)}^{†}+\frac{1}{4}Z^{1}X^{1}\left(\frac{1}{2}|0\rangle \langle 0|+\frac{1}{2}|1\rangle \langle 1|\right){\left(Z^{1}X^{1}\right)}^{†}\\ =\frac{1}{2}\left[\begin{array}{cc} 1 & 0\\ 0 & 1. \end{array}\right] \end{array}$$

These calculations indicate that all the states obtained by Calvin are just totally mixed states. So Calvin cannot learn Alice’s and Bob’s private binary strings from the particle sequences he obtained.

### 4.4. Comparison

The related quantum PSI-CA protocols in [27,28] required entangled states, other complicated oracle operators and measurements in high dimensional Hilbert space, hence it is more feasible with the current technologies than those proposed with entangled states. Compared with some recently proposed protocols [27,28], our proposed quantum PSI-CA protocol has the following advantages. First, it only needs to take single photons as quantum resources and to apply single operators and measurements. Obviously, it is more feasible to prepare these resources and implement these operators and measurements. Second, our new protocol is more robust and can easily use the fault tolerant technologies due to single photons. Therefore, our new quantum protocol for PSI-CA is more practical and feasible compared with the existing protocols.

## 5. Discussion and Conclusions

In summary, we give a novel quantum solution for PSI-CA. With the help of the quantum operators *X*, *Z*, and *T*, Calvin can help Alice and Bob obtain the PSI-CA results of their private sets after performing. Moreover, we provide a theoretical correctness study and use the Qiskit package to verify the scheme on the IBM Quantum Experience platform by way of a simulation experiment. In the end, we provide a security analysis of our protocol, which demonstrates that our protocol can resist various outside attacks, such as the disturbance attack, the Trojan horse attack, the intercept–resend attack, the entanglement-and-measure attack, and the man-in-the-middle attack. Additionally, it can also overcome the problem of information leakage with acceptable efficiency. Furthermore, we hope to extend our protocol for a generic case such as an *n*-qubit Toffoli gate and we also hope that our methods can provide some new ideas to solve more secure multi-party computations in the future.

## Figures and Tables

**Figure 1 entropy-25-00516-f001:**
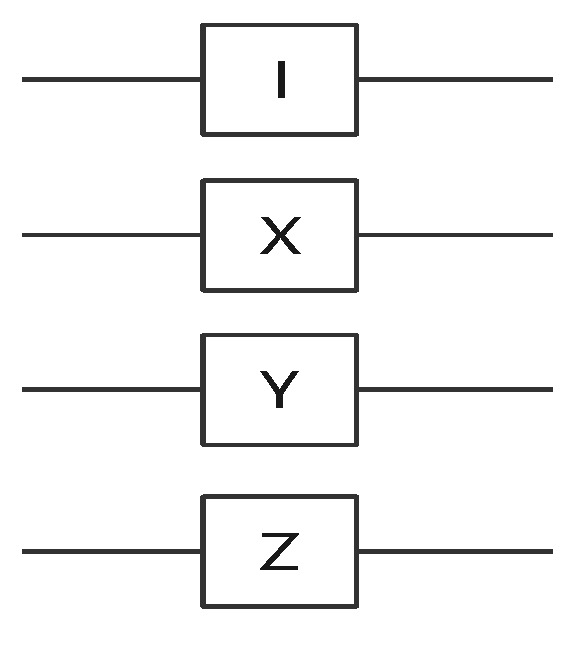
The circuit symbols for the four single-qubit gates, I,X,Y,Z.

**Figure 2 entropy-25-00516-f002:**
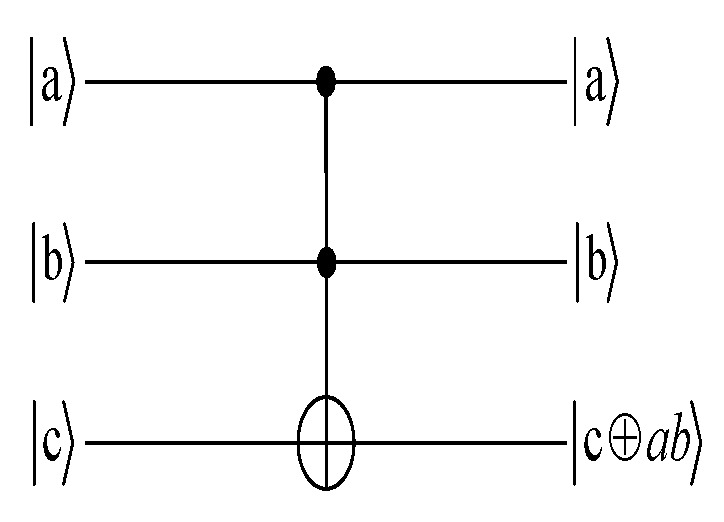
The circuit symbol for the *T* gate.

**Figure 3 entropy-25-00516-f003:**
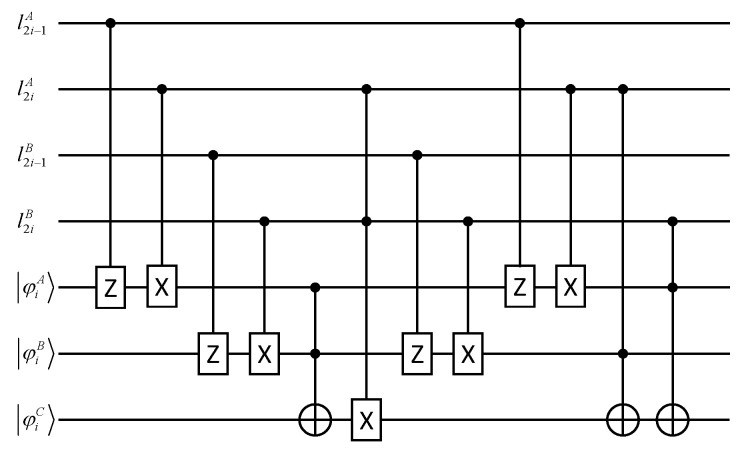
The circuit *U* used to privately calculate $T|\psi_{k}^{A}\rangle |\psi_{k}^{B}\rangle |\psi_{k}^{C}\rangle$.

**Figure 4 entropy-25-00516-f004:**
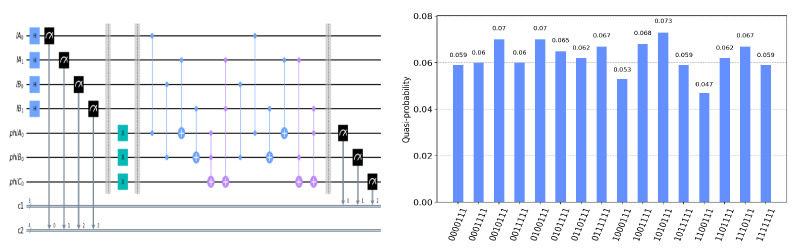
The circuit used to privately calculate $T|1\rangle |1\rangle |1\rangle$ and the experiment results.

**Figure 5 entropy-25-00516-f005:**
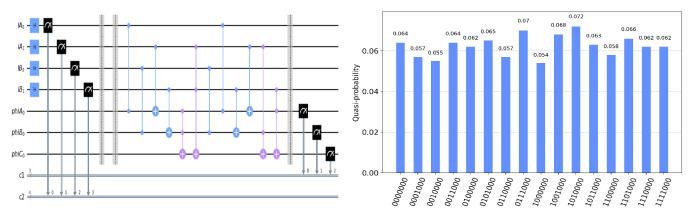
The circuit used to privately calculate $T|0\rangle |0\rangle |0\rangle$ and the experiment results.

**Figure 6 entropy-25-00516-f006:**
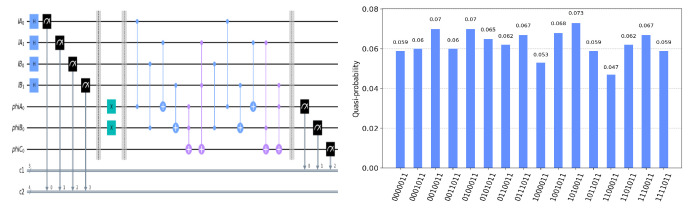
The circuit used to privately calculate $T|1\rangle |1\rangle |0\rangle$ and the experiment results.

**Figure 7 entropy-25-00516-f007:**
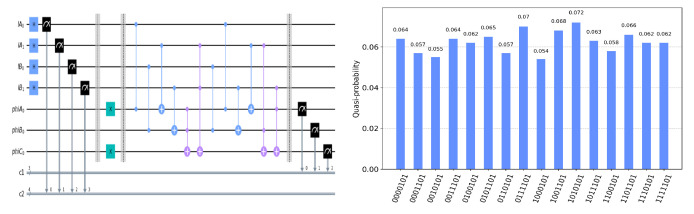
The circuit used to privately calculate $T|1\rangle |0\rangle |1\rangle$ and the experiment results.

**Figure 8 entropy-25-00516-f008:**
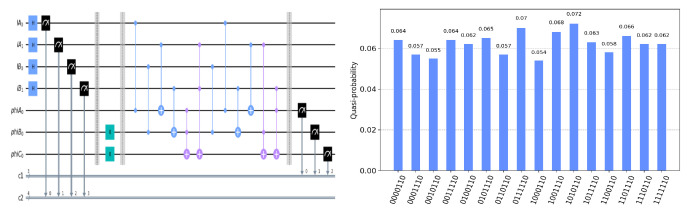
The circuit used to privately calculate $T|0\rangle |1\rangle |1\rangle$ and the experiment results.

**Figure 9 entropy-25-00516-f009:**
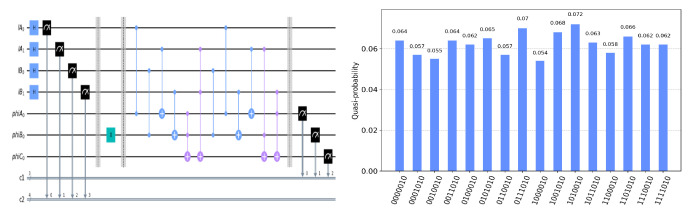
The circuit used to privately calculate $T|0\rangle |1\rangle |0\rangle$ and the experiment results.

**Figure 10 entropy-25-00516-f010:**
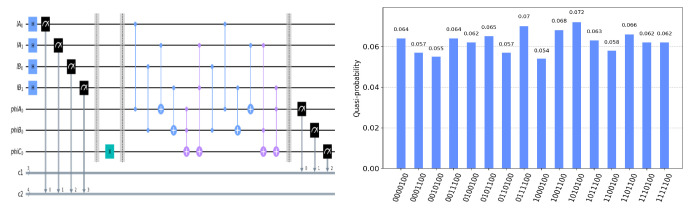
The circuit used to privately calculate $T|0\rangle |0\rangle |1\rangle$ and the experiment results.

**Figure 11 entropy-25-00516-f011:**
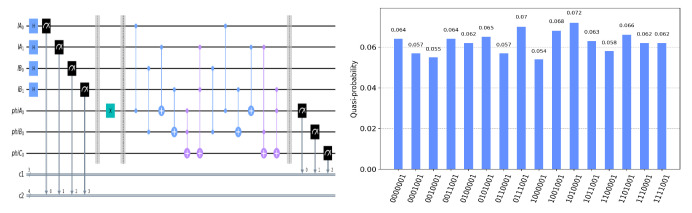
The circuit used to privately calculate $T|1\rangle |0\rangle |0\rangle$ and the experiment results.

## Data Availability

The data presented in this study are available on request from the author.

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
