# Peer review of "A New Quantum Private Protocol for Set Intersection Cardinality Based on a Quantum Homomorphic Encryption Scheme for Toffoli Gate"

_entropy, 2023, doi:10.3390/e25030516_

Round 1
Reviewer 1 Report
The authors proposed a new quantum private protocol for set intersection cardinality based on quantum homomorphic encryption for the Toffoli gate. The proposed protocol was verified and simulated on the IBM Quantum Experience platform. The specific process of each step of the scheme is introduced in detail. I recommend the authors to revise this manuscript by the comments below and then it can be accepted.
[1] The authors introduce a semi-honest third party without giving a specific description of the semi-honest model. The authors should give a specific introduction to the semi-honest model or directly cite relevant references.
[2] In the introduction and comparison of literature [16] and [17-18], only [16] is mentioned, and the introduction of [17-18] is lacked, so the conclusion seems abrupt.
[3] Sec. 4.1 proves the correctness of the algorithm. I suggest that the authors provide a simple and specific example, which may make the algorithm more explicit.
[4] In Section 3, the symbol definition of the protocol only introduces Alice and Bob, but the third party Calvin suddenly appeared in the second step of the protocol. The author should introduce the introduction of the third party when defining the objects and symbols of the protocol.
[5] In the second step of their protocol, Alice and Bob send the locations of random single photons to the third party through a QSDC protocol, but the authors did not introduce the QSDC protocol. The author should add a detailed introduction to the protocol, or give an example showing how to use the QSDC protocol.
[6] The authors should make a detailed comparison between their proposed algorithm and the existing ones. A table might better show the results of the comparison.
[7] When the authors say “The security of classical PSI-CA protocols is based on the computational complexity assumptions, which are vulnerable to attack by the quantum computer”, some references (e.g. Shor, Grover) should be given.
Reviewer 2 Report
- Specific comments
1. If authors can have some descriptions for each of their experimental results for Fig. 4 – Fig. 11, it would be great helps to readers who are new to the topic.
2. in Figure 3,
are not appeared in the description of the figure, a clarification is needed.
3. The authors made a good conclusion. However, a potential future exploration may make the proposed quantum protocol more valuable.
Please take a close look at the following typos:
o Page 1, line 20, y_i = {(x_i)
o Page 2, line 46, "he"-->"they"
o Eqs. (6) and (23) should be adjusted to align with the content.
On page 6, “According to equations (8)-(23),” à “According to equations (7)-(22),”

Round 2
Reviewer 1 Report
The authors have properly revised the manuscript which is now suitable for publication. I appreciated their explanation about the points raised in my previous report and I'm convinced that this manuscript contains relevant advancements for the field.
Reviewer 2 Report
I am satisfied with the revisions.